# Energy Storage Application of CaO/Graphite Nanocomposite Powder Obtained from Waste Eggshells and Used Lithium-Ion Batteries as a Sustainable Development Approach

**DOI:** 10.3390/nano14131129

**Published:** 2024-06-30

**Authors:** Kathalingam Adaikalam, Aviraj M. Teli, Karuppasamy Pandian Marimuthu, Sivalingam Ramesh, Hyungyil Lee, Heung Soo Kim, Hyun-Seok Kim

**Affiliations:** 1Millimeter-Wave Innovation Technology (MINT) Research Center, Dongguk University-Seoul, Seoul 04620, Republic of Korea; kathu@dongguk.edu; 2Division of Electronics and Electrical Engineering, Dongguk University-Seoul, Seoul 04620, Republic of Korea; avteli.teli@gmail.com; 3Department of Mechanical Engineering, Sogang University, Seoul 04107, Republic of Korea; pandian@sogang.ac.kr (K.P.M.);; 4Department of Mechanical, Robotics and Energy Engineering, Dongguk University-Seoul, Seoul 04620, Republic of Korea; sivaramesh1064@gamil.com (S.R.); heungsoo@dgu.edu (H.S.K.)

**Keywords:** chicken eggshell, CaO, graphite rods, waste utilization, supercapacitor, energy storage, sustainability

## Abstract

The reuse of waste materials has recently become appealing due to pollution and cost reduction factors. Using waste materials can reduce environmental pollution and product costs, thus promoting sustainability. Approximately 95% of calcium carbonate-containing waste eggshells end up in landfills, unused. These eggshells, a form of bio-waste, can be repurposed as catalytic electrode material for various applications, including supercapacitors, after being converted into CaO. Similarly, used waste battery electrode materials pose environmental hazards if not properly recycled. Various types of batteries, particularly lithium-ion batteries, are extensively used worldwide. The recycling of used lithium-ion batteries has become less important considering its low economic benefits. This necessitates finding alternative methods to recover and reuse the graphite rods of spent batteries. Therefore, this study reports the conversion of waste eggshell into calcium oxide by high-temperature calcination and extraction of nanographite from spent batteries for application in energy storage fields. Both CaO and CaO/graphite were characterized for their structural, morphological, and chemical compositions using XRD, SEM, TEM, and XPS techniques. The prepared CaO/graphite nanocomposite material was evaluated for its efficiency in electrochemical supercapacitor applications. CaO and its composite with graphite powder obtained from used lithium-ion batteries demonstrated improved performance compared to CaO alone for energy storage applications. Using these waste materials for electrochemical energy storage and conversion devices results in cheaper, greener, and sustainable processes. This approach not only aids in energy storage but also promotes sustainability through waste management by reducing landfills.

## 1. Introduction

Due to the continuous depletion of fossil fuels and associated environmental issues, the identification of new energy source materials that do not pollute the environment has recently gained attention for sustainability purposes [1]. Moreover, there is increasing research focusing on clean and renewable energy sources [2]. In this respect, solar and wind technologies are gaining attention, as they generate energy capable of replacing fossil fuels. However, efficient storage devices are required to make renewable power sources practical. In this case, supercapacitors and rechargeable batteries play a credible role in storing the electricity produced. Rechargeable batteries are expected to play significant roles in future portable energy storage devices useful for wearable and implantable devices [3,4]. Supercapacitors are very attractive for energy storage due to their high efficiency in storage, reliability, and tunable properties [5]. This electrochemical storage of electrical energy is efficient and eco-friendly. For electrochemical energy storage devices, such as supercapacitors, electrodes play major roles in the storage process [6]. 

Numerous research efforts are focused on finding new materials with efficient electrochemical performance for high-storage supercapacitors [7,8,9]. Natural resources are also being explored for these storage devices, which can lead to resource depletion and environmental issues. In addition to the devastation caused by fossil fuels, common practices of waste recycling or disposal further pollute the environment and contribute to economic weakness. Therefore, for sustainable growth and development, waste materials should be efficiently reused as secondary energy sources.

Solid waste materials have increased due to population growth and industrial revolutions [10]. These ever-increasing wastes are becoming a challenge to the global community and its sustainable development. Improper management of solid wastes affects public health and the environment. Some waste produced by human activities can be converted for reuse, fostering sustainable environments. Recently, the reuse of bio-waste and e-waste in the energy field has increased significantly [11]. To meet the increasing demand for future energy needs, low-cost, energy-efficient materials that can be easily prepared on a large scale are essential for energy storage applications. The materials used for the electrodes determine the properties, efficiency, and sustainability of the produced supercapacitors. In this context, sustainability refers to using waste materials as electrode materials for supercapacitor applications without compromising efficiency or causing damage to the environment and resources.

Oxides, phosphates, and their combinations are readily available at a low cost and by simple processing methods. However, most of these materials are not easily recyclable, thus requiring tedious processes and expensive instruments, which can cause serious problems. Therefore, it is crucial to find novel sustainable materials that are both renewable and efficient for developing next-generation technologies, including energy storage devices [12]. When reusing waste materials for sustainable development, the process should be inexpensive and not generate harmful materials, to maintain a clean environment. In this study, we selected two waste materials—waste chicken eggshells and used graphite rods—to explore their application in electrochemical supercapacitors after conversion. In this conversion of waste eggshells into CaO and used graphite rods into pure graphite or graphene particles, no chemicals were used, making it a clean and environmentally benign technique. 

Chicken eggshells are largely produced as waste by homes, restaurants, and some industries that use chicken eggs as raw materials. These eggshells are typically dumped as solid waste in landfills, causing pollution. The main component of eggshells is CaCO_3_, which comprises more than 90%. CaCO_3_ is an attractive divalent ion source and can be effective for energy source applications; hence, it has been explored for energy storage applications [13]. This eggshell CaCO_3_ can be easily converted into CaO through simple calcination, making it useful for various applications, including the agriculture, cement, and food industries [14,15,16,17]. However, the use of CaO derived from waste eggshells in scientific applications is relatively low. Some studies have explored using CaO obtained from eggshells through high-temperature calcination as an electrode for supercapacitor applications, although they are limited [18]. Like petroleum-based and other bio-waste [19], e-waste can be converted into carbon. Among the different types of e-waste, graphite rods obtained from used battery cells can be utilized to produce graphene or carbon-based materials [18]. These carbon-based materials obtained from waste materials play crucial roles in various applications, particularly in energy applications as electrode materials. Carbon-based materials are highly suitable for supercapacitor applications due to their low production cost, abundant availability, stability, good electrical conductivity, and large surface area with high power and energy densities. Graphite, another form of carbon, is globally recognized as a valuable resource material for various applications in the technological world due to its outstanding electrical, thermal, and mechanical properties [20]. It also serves as a source for different electronic and energy applications due to its different forms, such as graphene and other types of graphite, and as a source for carbon-based materials. Graphite has been extensively used across different industries for various applications, and its demand is expected to increase as an electrode material for future battery applications. It is an attractive anode material that is universally used for lithium-ion batteries and other electrochemical applications due to its outstanding features [21,22].

Most lithium-ion battery manufacturers utilize natural graphite as anode materials, which subsequently becomes waste after their use [23]. These spent graphite rods can serve as a potential source of graphite and graphene-based materials [24]. Therefore, the reuse of waste graphite rods is gaining attention for electrochemical applications [25]. Specifically, they are often discarded as normal waste by the common man, significantly polluting the environment and soil. For sustainable development, the recovery and reuse of discarded graphite electrodes are essential [26]. There is no standard recycling procedure for discarded graphite anodes after usage; most recycling techniques are expensive. Thus, this study reports the conversion of waste eggshell and spent graphite rods into CaO and graphite particles, respectively, and their use for supercapacitor application. Recycled graphite finds applications in various fields, as it provides porous carbon materials suitable for high efficient current collecting electrodes [3,20]. However, this is the first study to use recycled graphite for supercapacitor application and as a composite with bio-waste-derived materials. Characterization of converted materials and their respective conversion process are also reported. 

## 2. Experimental Details

All the chemicals used in this study were purchased from Sigma Aldrich Pvt. Ltd. (Seoul, Republic of Korea) and were used as received. Waste chicken eggshells collected from kitchen waste were crushed into small pieces and washed several times with deionized (DI) water to remove unwanted residues. The washed eggshells were then dried at room temperature for two days. After drying, the eggshells were ground into a fine powder using a mixer grinder. This eggshell powder was then calcined at approximately 900 °C for 1 h. Finally, the calcined powder was cooled to around 100 °C, and the resulting CaO powder was crushed using a pestle and mortar to obtain a fine powder with nanoscale particle size.

For graphite powder preparation, a waste graphite rod was collected from a used lithium-ion battery, and part of the outer layer was manually removed using a knife and sandpaper. The remaining rod was then slowly scratched and crushed into powder. The resulting powder was leached using ammonia, ammonium sulfite, and DI water. For leaching, 10 g of graphite powder was added to 100 mL of leaching agent in a conical flask and stirred for 1 h at 50 °C. After the leaching process, the graphite powder was filtered and washed several times, then dried. Finally, the graphite powder was annealed at 200 °C for 1 h to remove moisture and other organic compounds. The typical experimental procedure followed for the preparation of eggshell-derived CaO and graphite powder from used graphite rods is schematically shown in Figure 1.

XRD measurements were performed using a Bruker Discover D8 diffractometer (Billerica, MA, USA) with a CuKα radiation source (λ = 1.5418 A). A scanning electron microscope (SEM, Hitachi S-4800, Tokyo, Japan) was used to obtain surface morphological and compositional properties. The surficial chemical composition of the composites and their oxidation states were analyzed using X-ray photoelectron spectroscopy (XPS; Versaprobe II, ULVAC-PHI Inc., Chigasaki, Kanagawa, Japan). Furthermore, their potential as electrode materials for supercapacitor applications was analyzed using a WonAtech ZIVE-SP5 (Seoul, Republic of Korea) electrochemical workstation. The electrochemical studies, such as cyclic voltammogram (CV), galvanostatic charge–discharge (GCD), and electrochemical impedance spectroscopy (EIS), were used to analyze the electrochemical properties of the composite materials. For electrochemical analysis, a standard three-electrode cell setup with Pt and Ag/AgCl electrodes were used as the counter electrode and reference electrode, respectively, with prepared CaO/graphite nanocomposite powder as the working electrode in 2 M KOH aqueous solution. The prepared nanocomposite material was made as a paste by mixing it with PVDF and carbon black in the mass ratio 70:10:20 inN-methyl-2-pyrrolidone. The produced slurry of around 2 mg was applied onto nickel foam of 1 cm^2^ area to prepare the electrode for the electrochemical studies. The nickel foam was treated with 3 M HCl before slurry coating, then dried in a vacuum for 12 h at 80 °C. Specific capacitance (*C_s_*) values of the materials were estimated using Equation (1).
$$C_{s}=\frac{2}{mV^{2}}\int iV\left(t\right)dt$$
where *t* is the discharge time, *m* is the mass of active material, *i* is the galvanostatic discharge current, and $\int V\left(t\right)dt$ is the area under the galvanostatic discharge curve [27,28]. 

## 3. Results and Analysis

### 3.1. Structural and Morphological Analysis

The morphological and structural properties of the converted CaO, graphite, and CaO/graphite nanocomposites were analyzed using SEM, TEM, and XRD techniques. Waste eggshells were converted into crystalline pure CaO nanopowder through simple calcination at approximately 900 °C. At this high temperature, the major constituents of eggshell CaCO_3_ and Ca(OH)_2_ decompose with the evaporation of water molecules, causing conversion into CaO nanoparticles (NPs). The obtained CaO NPs and CaO/graphite nanocomposites were characterized to analyze their structural and morphological properties using X-ray diffraction studies. Figure 2a shows the XRD patterns of calcined eggshell, displaying peaks at 32.3°, 37.4°, 54°, 64.2°, 67.5°, and 79.8°, which correspond to the (111), (200), (220), (311), (222), and (400) planes of CaO NPs, respectively (PDF Card No. 99-0070). The sharp peaks with high intensity indicate the formation of highly crystalline polycrystalline CaO NPs [10]. The peak at 37.4° shows an interplanar distance of 0.475 nm, matching with reported values of crystalline cubic CaO (space group: Fm-3 m) [29]. This interplanar distance was calculated using Bragg’s law, nλ=2dsin θ, where *n* is the order of diffraction (*n* = 1), *λ* = 1:54 A° wavelength of X-rays, and *θ* is the diffraction angle of the XRD peak. This confirms the complete conversion of eggshells into crystalline CaO particles. The XRD pattern of the CaO/graphite composite is shown in Figure 2b. It displays peaks at 26. 48°, 28.69°, 31.50°, 34°, 37.4°, 43.45°, 50.8°, 54.37°, 59.6°, 64.3°, and 72°, confirming the inclusion of graphite powder with CaO. Peaks located at 31.50°, 37.4°, and 64.3° indicate the inclusion of CaO with graphite flakes. The other peaks are attributed to the included graphite and its related carbon peaks. Peaks observed at 28.69°, 34°, and 43.45° are respectively attributed to the (002), (020), and (201) planes of crystalline carbon [11]. The peaks at 26.48°, 50.80°, 54.37°, 59.65°, and 72.06° correspond to (002), (101), (004), (110), and (112) planes of crystalline graphite platelet NPs, respectively [30]. The strong diffraction peak exhibited at 26.48° indicates the presence of graphite in the composite [31]. The interplanar distance of 0.346 nm, calculated using the peak (002), indicates distinguishable graphite particles. These observations infer that the obtained graphite from waste graphite rod is a highly oriented carbon-based material [32].

FE-SEM images obtained to analyze the surface morphology of the prepared nanocomposite are displayed in Figure 3. Figure 3a–c show the SEM images of the prepared CaO NPs, displaying multi-shaped NPs of different sizes. The SEM images shown in Figure 3d–f depict the composite of CaO/graphite powder, which indicates different sizes and shapes of nanoplatelet morphology [15]. Most of the particles have a layered nanoplate structure. Both CaO and graphite particles exhibit layered structures, with CaO having more regularly shaped cylindrical rods than platelets. In contrast, the pure graphite particles displayed in Figure 3g,h appear as crystalline, flake-like smooth surfaces of graphite nanocrystals. The irregular arrangement of the different-sized graphite flakes forms rough surfaces with interlayer cavities and pores. Low-magnification SEM images of CaO and graphite nanopowders are shown in Appendix A to differentiate the morphologies of the two different particles. 

TEM is valuable for analyzing the microstructural properties of nanocomposite materials. Therefore, the structural properties of the prepared CaO and CaO/graphite composites were examined using a high-resolution field emission transmission electron microscope (HR FE-TEM). Figure 4 shows TEM images of CaO samples at different magnifications with the SAED pattern and interplanar distance calculated using Gatan digital micrograph software (AMETEC, Berwyn, PA, USA). The images display the plate-like morphology of CaO particles of varying sizes. The SAED pattern obtained shows spot-like features, indicating the crystalline nature of the prepared CaO NPs. The high-resolution image presents fringe patterns, further confirming the crystalline nature of the CaO NPs. The interplanar distance of 0.471 nm calculated from this TEM image coincides with XRD results. Similarly, TEM images of CaO/graphite composites are shown in Figure 5. The images display particles of different sizes. Compared to CaO, the CaO/graphite exhibits a nanoflake-like structure due to the graphite layers derived from used graphite rods. This flake-like morphology results from the slicing of graphite during the grinding process. The high-resolution image presents different fringe patterns, indicating the composition of different crystalline phases of the material. The SAED pattern obtained shows spot-like features with rings, indicating the composite nature of CaO/graphite powder with the crystalline and amorphous nature of the nanocomposites. The interplanar distance obtained from a spot in the SAED pattern employing Gatan software was approximately 0.349 nm, further confirming the graphitic phases of the material. These observations are consistent with XRD results.

### 3.2. Chemical Compositional Analysis 

Energy dispersive X-ray analysis (EDAX) was also performed to determine the chemical composition of the composites. The EDAX spectra of the CaO, graphite, and CaO/graphite composite NPs are shown in Figure 6. The appearance of Ca and O peaks for the CaO sample (Figure 6a) confirms the formation of CaO in this calcination process, as inferred from XRD results. In contrast, the CaO/graphite composite powder shows C peaks along with Ca and O, confirming the inclusion of graphite with CaO (Figure 6b). This is further validated by the EDAX result of pure graphite powder (Figure 6c), which shows only a carbon peak, indicating that the graphite plates obtained from used graphite rods are almost pure carbon without any impurity elements.

The use of different kinds of materials like oxides and nanocarbon-based materials as complex composites creates unpredictable outcomes due to the nature of surficial interactions between the two different materials. The information about elemental compositions and their related surficial reactions can be obtained using XPS studies [33]. Hence, XPS was conducted on the prepared CaO/graphite nanocomposite to identify the elemental composition and the oxidation states of the elements. The obtained spectra were analyzed by Gaussian fitting using Fityk 1.3.1 software with background corrections. The XPS survey spectrum of the composite, shown in Figure 7a, displays Ca 2s, Ca 2p, O 1s, and C 1s peaks, indicating the presence of the required elements Ca, O, and C in the prepared composite. The deconvoluted high-resolution spectra of Ca 2p, O 1s, and C 1s obtained using XPS recording with a pass energy 29.25 eV at 0.125 eV step range are shown in Figure 7b, Figure 7c, and Figure 7d, respectively. The binding energy values of the elements are consistent with the reported values [2,31]. The spin–orbit doublet peaks, 2p_3/2_ and 2p_1/2_, of the Ca 2p spectrum exhibit a separation of 3.65 eV. This separation energy of the Ca^2+^ oxidation state is slightly higher than that of pure CaO (i.e., 3.5 eV) due to the interaction of graphite in the composite of the calcined chicken eggshell electrode for battery and supercapacitor applications. Figure 7c shows the deconvoluted peaks of the O 1s spectrum, which exhibits two peaks at approximately 530 eV and 531 eV, associated with the oxygen in CaO and other oxygen species found in the composite, respectively.

The C 1s spectrum shows asymmetric and complex structure, indicating the interaction between CaO and graphite, and it is well resolved by deconvolution.

Figure 7d shows the fitted C 1s spectra with peaks at 282.7 eV, 283.2 eV, 283.45 eV, 286.7 eV and 288.6 eV, assigned to C–H, C–C, C=O, O=C–O and C–O vibrations, respectively, due to the association of carbon with CaO [31]. The main peak of the deconvoluted spectrum, located at 283.2 eV, is in an asymmetric shape from the higher binding energy side; it confirms the C-C graphene phase of sp^2^ hybridization as inferred form Doniach-Sunjic function [33,34,35,36]. Chemisorbed oxygen from solvents on the graphite surface is attributed to the effect of C-O (286.7) eV. It also shows a graphite-induced C=C peak and other C-H, C=C-O peaks. All the peaks are little decreased from standard values; this may be caused by the interaction of formed surficial solvent-induced species [4]. The solvent-induced hydrogenation can also cause asymmetric peaks and extra bonds for graphene and graphite layers [37]. This oxide- and carbon-based heterogeneous catalyst can have multiple chemical states, causing asymmetric signals with varying binding energies in XPS results [38]. Particularly, the case of graphene-like layer material hydrogenation is playing vital roles in XPS results [37,39]. Along with XRD and TEM, this XPS result also confirms the combination of CaO and graphite in the prepared composite material.

### 3.3. Electrochemical Characterizations

We studied the electrochemical performance of CaO NPs and their composite with waste battery-derived graphite particles for energy storage applications. To analyze the electrochemical properties of the prepared materials, cyclic voltammetry (CV), galvanostatic charge–discharge (GCD), and electrochemical impedance spectroscopy (EIS) were performed using a three-electrode electrochemical cell setup with 2 M KOH aqueous solution as the electrolyte. Figure 8a shows the CV curves of CaO NPs recorded in the voltage range from 0 to 0.5 V vs. SCE at different high and low scan rates, ranging from 10 to 100 mV/s (Figure 8a) and 1 to 5 mV/s (Figure 8b). The CV curves display well-aligned characteristic loops, indicating distinct redox peaks for all the scan rates. This signifies the stable performance of the electrode with good rate capability and a reversible Faradaic redox reaction, indicating the rapid and reversible charge transfer characteristics. The scan rate-dependent increase in redox peaks indicates a steady and fast redox reaction between the electrolyte and electrode surface. The rectangular-shaped CV curves with well-defined symmetrical redox peaks suggest battery-type storage properties of the materials. This CV nature confirms the hybrid storage mechanism of the device, which includes both Faradaic behavior and electric double layer (EDLC) mechanisms. The scan rate-dependent changes in reduction and oxidation peaks reveal that at high scan rates, the CV curves display small double peaks, while at low scan rates, double peaks appear in the oxidation states. This confirms the mixed state of pseudocapacitive and Faradaic redox reactions. Figure 8d shows the CV curves of the CaO/graphite electrode recorded at scan rates of 1–5 mV/s. It also depicts the same for the CaO electrode, with slight differences in oxidation peaks and current levels. These distinct redox peaks at all scan rates demonstrate a high degree of reversible Faradaic reaction. Both samples showed the same CV nature, with slight differences in reduction and oxidation peaks with increased current, as shown in the comparative CVs in Figure 8f. The linear change in current with scan rate indicates the Faradaic nature of the reaction in both electrodes.

The energy storage mechanism and related kinetics were analyzed using CV plots of the materials recorded at different scan rates, employing Equation (1) derived from Dunn’s method [40]:(1)$$i=k_{i}+k_{2}\nu^{\frac{1}{2}}$$
where *i* and *ν* represent the peak current in mA and scanning rate in mV/s, respectively, and *k*_1_ and *k*_2_ are constants. The electrode contribution in capacitive and diffusion processes is calculated using Equation (1). The calculated capacitive and diffusive contribution of the CaO electrode’s current are illustrated in Figure 8c. As shown in this figure, the capacitive contribution is higher, approximately 80%, at lower scan rates compared to the diffusive contribution, and it slowly increases with the increase in scan rates. The increase in diffusive contribution with scan rate indicates a surface-controlled mechanism of the electrode [28]. In contrast, the CaO/graphite composite yielded a more diffusive contribution (78.7%) compared to the capacitive contribution, as shown in Figure 8e. This confirms that platelet-structured porous graphite sheet incorporation with CaO NPs contributes to increased diffusion. This diffusion contribution prominence is good for the electrode material’s high storage capacity. 

The GCD curves of the two electrodes, recorded at 0.3 mA for a potential window of 0–0.5 V vs. SCE, are shown in Figure 9a. These curves compare the performance of the electrodes during the electrochemical charging and discharging process. Both electrodes exhibited similar curve shapes, with the CaO/graphite electrode showing increased charge and discharge times, indicating its enhanced activity. The plateau-like GCD curves for both electrodes suggest the presence of active sites and excellent reversibility, characteristic of battery-type behavior [28]. Both CaO and CaO/graphite electrodes exhibited nonlinear, triangular-shaped symmetric curves, indicating a good charge–discharge process with high reversibility and coulombic efficiency, as well as minimal polarization. Thus, the CV and GCD results indicate the excellent capacity behavior of the material with battery-type behavior. The specific capacity, areal capacitance, energy density, and power density of the electrodes are provided in Table 1.

The cyclic stability of the CaO/graphite electrode was evaluated to assess the material’s capacity retention. The electrode was examined at 7 mAcm^−2^ for 50,000 cycles in the potential range of 0–0.5 V vs. SCE. As shown in Figure 9b, the CaO/graphite electrode maintains a specific capacity of approximately 90% after 50,000 cycles of rapid charge and discharge. Initially, the capacitance of the electrode increased to 123.7% for up to 20,000 cycles, and after that, it decreased to approximately 90% after 50,000 cycles, indicating the good cyclic stability and reversibility of the material [41]. The cyclic stability of CaO NPs for capacity retention was also tested for 30,000 cycles of charging and discharging, and it is shown in Appendix A. As seen from the figure, it also has reasonably good stability above 80% after 30,000 cycles of charging and discharging. Compared with pure CaO, the CaO/graphite composite material showed improved performance, increasing uniformly up to 30,000 cycles. This justifies the performance of graphite plates incorporated with CaO NPs.

EIS was conducted to analyze the charge storage behavior and the resistive and capacitive properties of the electrodes. Figure 9c displays the Nyquist (EIS) plots obtained for the prepared samples, which depict a straight-line nature in the low-frequency region with a slight curvature in the high-frequency region. This small curvature indicates high transfer resistance, whereas the equivalent series resistance (ESR) obtained from the straight-line part is very low, as shown in the inset of Figure 9c. The absence of semi-circles in the high-frequency region indicates excellent ion transfer properties of the composites. Thus, the Nyquist plots illustrate the effects of the series and Warburg resistance of the device. The CaO/graphite sample exhibits a smaller semicircle region with a shorter straight line compared to pure CaO. This indicates lower contact resistance showing higher conductivity and lower diffusion resistance for the redox process. These findings confirm that the inclusion of graphite particles with CaO helps to enhance the electrical conductivity and charge storage properties of the electrode. Based on this electrochemical analysis, we conclude that the inclusion of graphite with CaO can synergistically improve electrochemical properties.

## 4. Conclusions

In this study, we successfully prepared a CaO/graphite nanocomposite from waste eggshells and graphite rods of used batteries using inexpensive and facile methods. We studied its electrochemical energy storage properties for supercapacitor applications. The structural, morphological, and chemical properties of the converted materials were analyzed using XRD, SEM, TEM, and XPS characterization. The results indicated efficient and clear conversion of waste eggshells into CaO and waste graphite rods into powder of graphite nanoflakes. Electrochemical studies performed on CaO and its composite with graphite powder showed encouraging effects for electrochemical energy storage. The addition of graphite powder improved the storage property of the CaO/graphite composite by providing more reaction sites and surface area. The composite retained approximately 90% of its maximum storage capacity even after 50,000 cycles of charge and discharge processes. As an outcome, this study demonstrates the potential of recycling waste materials for energy applications, contributing to sustainable development and reducing environmental pollution.

## Figures and Tables

**Figure 1 nanomaterials-14-01129-f001:**
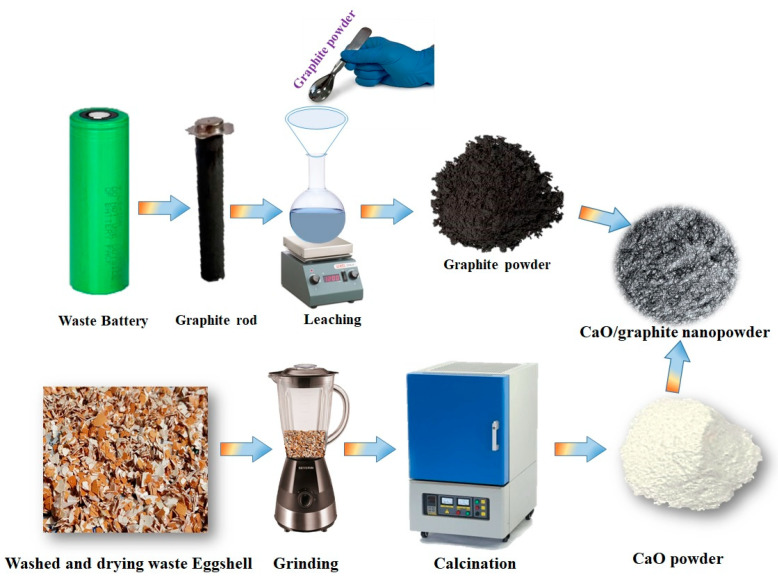
Schematic representation of process flow for the conversion of waste eggshell and used battery cell.

**Figure 2 nanomaterials-14-01129-f002:**
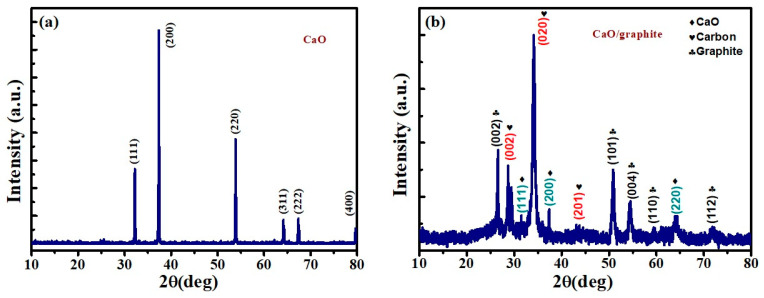
X-ray diffraction patterns of (**a**) CaO NPs and (**b**) CaO/graphite nanocomposite.

**Figure 3 nanomaterials-14-01129-f003:**
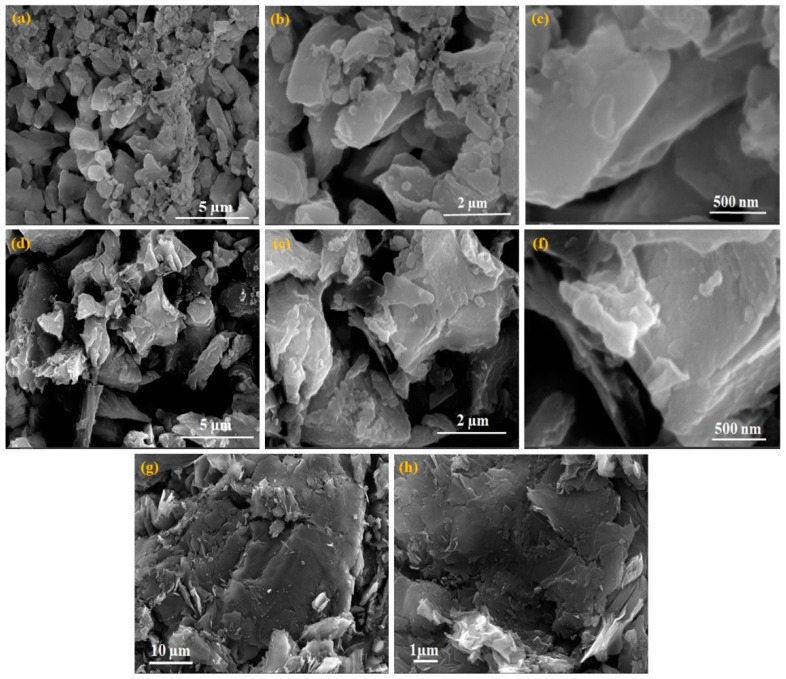
(**a**–**c**) SEM images of CaO NPs, (**d**–**f**) CaO/graphite nanocomposite, and (**g**,**h**) graphite powder.

**Figure 4 nanomaterials-14-01129-f004:**
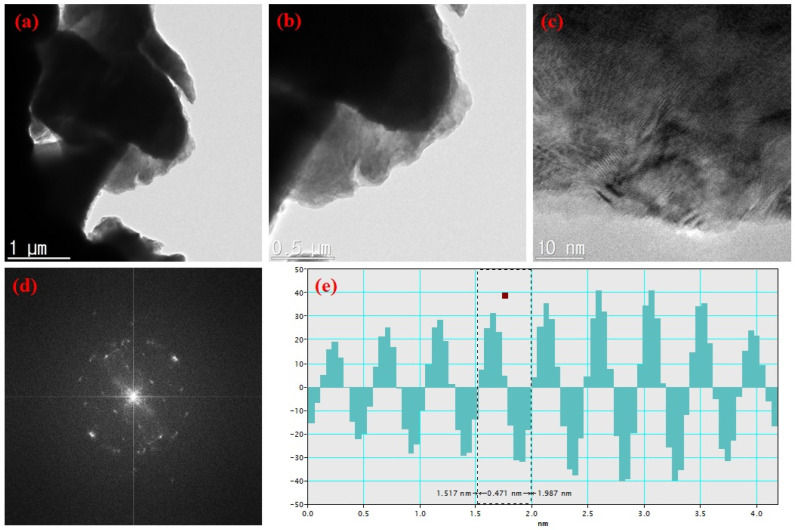
(**a**–**c**) TEM images of CaO NPs at different magnifications, (**d**) SAED pattern, and (**e**) interplanar distance.

**Figure 5 nanomaterials-14-01129-f005:**
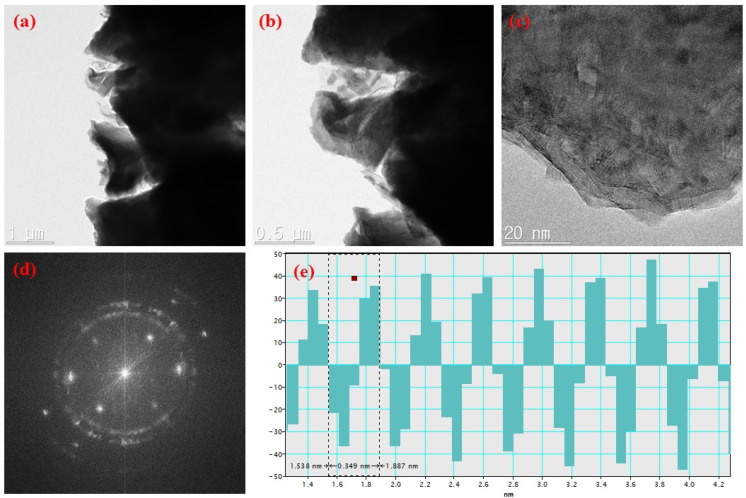
(**a**–**c**) TEM images of CaO/graphite nanocomposite at different magnifications, (**d**) SAED pattern, and (**e**) interplanar distance.

**Figure 6 nanomaterials-14-01129-f006:**
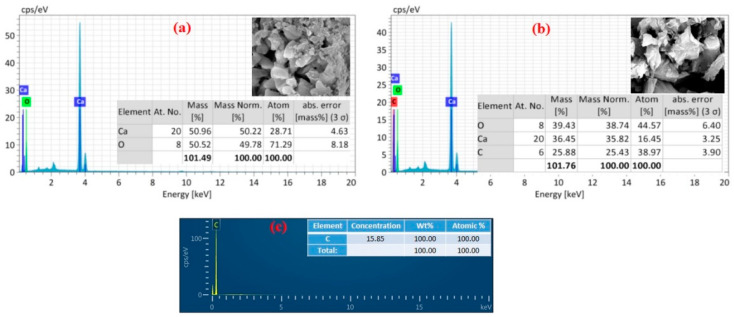
EDAX spectra and elemental composition of (**a**) CaO NPs, (**b**) CaO/graphite nanocomposite, and (**c**) graphite powder.

**Figure 7 nanomaterials-14-01129-f007:**
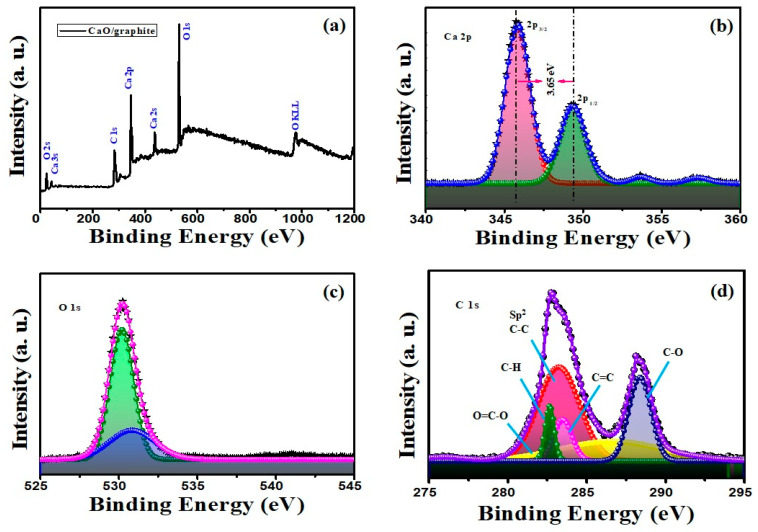
XPS spectra of CaO/graphite nanocomposite: (**a**) full survey, (**b**) Ca 2p, (**c**) O 1s, and (**d**) C 1s.

**Figure 8 nanomaterials-14-01129-f008:**
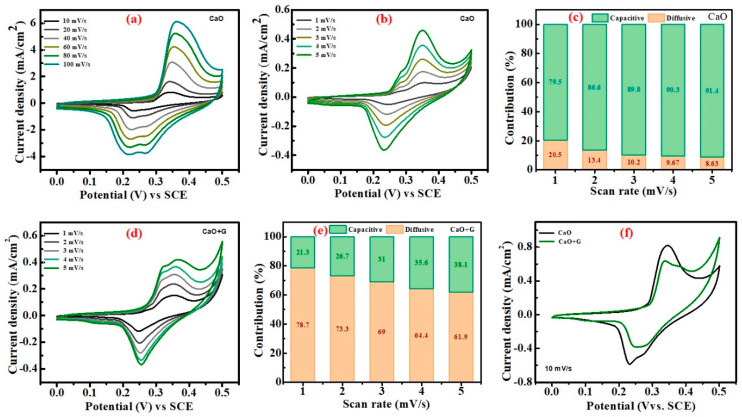
(**a**,**b**) CV curves of CaO at different scan rates, (**c**) capacitive and diffusion contributions CaO at different scan rates, (**d**) CV curves of CaO/graphite composite, (**e**) capacitive and diffusion contributions CaO/graphite at different scan rates, and (**f**) comparison of CV curves of CaO and CaO/graphite.

**Figure 9 nanomaterials-14-01129-f009:**
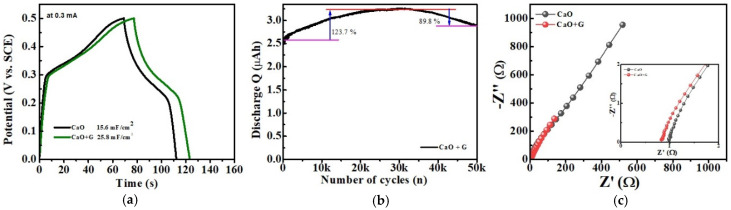
(**a**) GCD curves of CaO and CaO/graphite, (**b**) capacity retention vs. number of cycles plot, and (**c**) Nyquist plots (inset: enlarged Nyquist plots for the high-frequency region).

**Table 1 nanomaterials-14-01129-t001:** Electrochemical storage parameters of the electrode materials.

Sample Code	Areal Capacitance (C_A_; mF cm^−2^)	Specific Capacity (C; mAh cm^−2^)	Energy Density (ED; μWh cm^−2^)	Power Density (PD; μW cm^−2^)
CaO	25.8	12.9	0.45	75
CaO + G	27.6	13.8	0.48	75

## Data Availability

The data presented in this study are available on request from the corresponding author due to ethical reasons.

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
