# Peer review of "Energy Storage Application of CaO/Graphite Nanocomposite Powder Obtained from Waste Eggshells and Used Lithium-Ion Batteries as a Sustainable Development Approach"

_nanomaterials, 2024, doi:10.3390/nano14131129_

Round 1

Reviewer 1 Report

Comments and Suggestions for Authors

In order to reduce environmental pollution and cost consumption, this study reports the conversion of waste eggshell into calcium oxide by high-temperature calcination and extraction of nanographite from spent batteries for application in energy storage fields. The relevant measurements including XRD, SEM, TEM, XPS, and electrochemical tests also conducted. Overall, the research content of this work is innovative and practical. The manuscript provides a detailed description of the experimental methodology and data processing, and the experimental results are relatively reliable. The manuscript is acceptable for publication with significant revisions. The following are some specific suggestions:

Q1: In Introduction, as mentioned “supercapacitors and rechargeable batteries” (L.43), their relevant work could be focused: https://doi.org/10.1002/adfm.202312664 ; https://doi.org/10.1002/aenm.202302261. In addition, more sufficient literatures should be added in Introduction so as to enrich the text, e.g., L.40-42, L.69-71, etc.

Q2: In section 2, the CaO NPs were prepared by the carbonization of waste eggshells at approximately 900 ℃, why choose this temperature. Besides, whether the high-energy consumption of the carbonization process deviates from the theme of sustainable development approach for this work?

Q3: In Fig. 2d-f, which is graphite or CaO NPs? Suggest support the lower-magnification SEM images.

Q4: In Fig. 4, how to calculate the interplanar distance of CaO NPs with 0.471 nm?  More detailed description could be supplemented.

Q5: In Fig. 9b, the data about the cycling performance of CaO NPs should be supplied.

Author Response

Revision details-Author Response

Journal: Nanomaterials

Manuscript Title: Energy Storage Application of CaO/Graphite Nanocomposite Powder obtained from Waste Eggshells and Used Lithium-ion Batteries as a Sustainable Development Approach

Manuscript ID: nanomaterials-3079115

Authors would like to thank the Editor and Reviewers for their insightful queries and suggestions. Based on the reviewer queries the manuscript has been revised incorporating all the required contents. The author responses are highlighted in red color   both in author’s response and manuscripts.

Reviewer-1

In order to reduce environmental pollution and cost consumption, this study reports the conversion of waste eggshell into calcium oxide by high-temperature calcination and extraction of nanographite from spent batteries for application in energy storage fields. The relevant measurements including XRD, SEM, TEM, XPS, and electrochemical tests also conducted. Overall, the research content of this work is innovative and practical. The manuscript provides a detailed description of the experimental methodology and data processing, and the experimental results are relatively reliable. The manuscript is acceptable for publication with significant revisions. The following are some specific suggestions:

Reviewer query

Q1: In Introduction, as mentioned “supercapacitors and rechargeable batteries” (L.43), their relevant work could be focused: https://doi.org/10.1002/adfm.202312664 ; https://doi.org/10.1002/aenm.202302261. In addition, more sufficient literatures should be added in Introduction so as to enrich the text, e.g., L.40-42, L.69-71, etc.

[Lightweight Zn-Philic 3D-Cu Scaffold for Customizable Zinc Ion Batteries, Interrelation Between External Pressure, SEI Structure, and Electrodeposit Morphology in an Anode-Free Lithium Metal Battery]

Author Response

As suggested the introduction part is modified adding more points referring the given works.

Reviewer query

Q2: In section 2, the CaO NPs were prepared by the carbonization of waste eggshells at approximately 900 ℃, why choose this temperature. Besides, whether the high-energy consumption of the carbonization process deviates from the theme of sustainable development approach for this work?

Author Response

Yes, agree with reviewer’s view in this respect of sustainability. Just we referred earlier report for the conversion of Eggshell. As reviewer told, some alternative methods or eggshell’s CaCO3 itself after low temperature treatment without converting it as a pure CaO can be studied to analyze its effect in various fields. We are also interested, will try in future. 

Reviewer query

Q3: In Fig. 2d-f, which is graphite or CaO NPs? Suggest support the lower-magnification SEM images.

Author Response

Low magnification SEM images of CaO and graphite powder are provided in the supplementary materials part. The materials of the SEM images are clearly mentioned in the figure’s caption and the text as mentioned below. Figures 3(a–c) show the SEM images of the prepared CaO NPs, displaying multi-shaped NPs of different sizes. Figure 3. (a-c) SEM images of CaO NPs, (d-f) CaO/graphite nanocomposite, and (g, h) graphite powder. (the first three figures (a,b,and c) are for CaO at different magnifications and similarly for others.

Reviewer query

Q4: In Fig. 4, how to calculate the interplanar distance of CaO NPs with 0.471 nm?  More detailed description could be supplemented.

Author Response

At first, the interplaner distance (d) (distance between adjacent crystalline CaO planes) was calculated using XRD data applying Bragg’s law, ie.  , where n is the order of diffraction (n = 1), λ = 1:54 A° wavelength of X-rays and and θ is the diffraction angle of a particular peak. For the calculation of interplaner distance using TEM images Gatan software was used, in which SAED pattern was used to calculate the width of the fringe pattern which is equal to interplaner distance of the particular plane selected.

Reviewer query

Q5: In Fig. 9b, the data about the cycling performance of CaO NPs should be supplied.

Author Response

As suggested we have provided cycling stability of CaO NPs in the revised manuscript

Reviewer 2 Report

Comments and Suggestions for Authors

Reviewer report on manuscript nanomaterials-3079115

Kathalingam Adaikalam et al.Energy Storage Application of CaO/Graphite Nanocomposite Powder obtained from Waste Eggshells and Used Lithium-ion Batteries as a Sustainable Development Approach

This study reports the conversion of waste eggshell into calcium oxide by high-temperature calcination and extraction of nanographite from spent batteries for application in energy storage fields. Both CaO and CaO/graphite were characterized for their structural, morphological, and chemical compositions using XRD, SEM, TEM, and XPS techniques. The prepared CaO/graphite nanocomposite material was evaluated for its efficiency in electrochemical supercapacitor applications. CaO and its composite with graphite powder obtained from used lithium-ion batteries demonstrated improved performance compared to CaO alone for energy storage applications.   

The manuscript can be accepted only after major revision.

Below, I point out several questions to help the authors improve the manuscript before publication.

Questions/comments:

1.      The introduction doesn’t provide sufficient background and doesn’t include all relevant references, e.g.  [Yalovega et al. Interfacial interaction in NiOx/MWNTs and CuOx/MWNTs nanostructures as efficient electrode materials for high-performance supercapacitors. Nanomaterials, 2024, 14(11), 947] and [Shmatko et al. Interaction between NiOx and MWNT in NiOx/MWNTs composite: XANES and XPS study. Journal of Electron Spectroscopy and Related Phenomena, 2017, 220, 76-80].

2.      More details to the section “2. Experimental details” should be added.

3.      Detailed information about X-ray photoelectron spectra recording should be added. The energy resolution and pass energy values during X-ray photoelectron spectra acquisition should be written. The procedures for background subtraction and spectra deconvolution procedure should be well described.

4.      It is well-known that sp2 component of the X-ray photoelectron spectra has an asymmetric shape from the side of higher binding energies, which is described by the Doniach-Sunjic function, which should be used for fitting of the sp2 component in the C1s spectrum. As an example, I recommend using the publication [J. Alloys Compd. 2019, 792, 713-720].

5.      The XPS spectra fitting is not very good justified. There are not up-to-date references for choice of the components. I recommend using the publications [Small, 2023, 19, 2208265] and [Appl. Surf. Sci. 2022, 590, 153055].

6.      There are errors in the interpretation of peaks in the X-ray photoelectron spectra that must be corrected before the manuscript publication. I recommend referring to the publications [Small, 2023, 19, 2208265], [Appl. Surf. Sci. 2022, 590, 153055], and [Carbon, 2022, 194, 52-61].

7.      The statement “Figure 7(d) shows the fitted C 1s spectra with peaks at 285 eV, 286 eV, and 288 eV, assigned to C–C, C–O, and C=O vibrations, respectively, due to the association of carbon with CaO” (page 8) is not correctly for X-ray photoelectron spectra and must be corrected. For corrections, please, refer to the mentioned above publications.

8.      It’s not clear how Authors made the conclusion “This XPS result also confirms the combination of CaO and graphite in the prepared composite material.” (page 8). This discussion should be extended.

9.      Section “Funding” is missed.

10.  Section “Author Contributions” is missed.

11.  “Data availability statement” is missed.

12.  “Acknowledgments” is missed.

13.  “Statement about conflict of interest” is missed.

References should be corrected and reformatted. Almost all references have missing title of journals, e.g.  [1], [2], [7], [14-16], [19], [20], [22], [23], [25-29], and [31-33].

Author Response

Revision details-Author Response

Journal: Nanomaterials

Manuscript Title: Energy Storage Application of CaO/Graphite Nanocomposite Powder obtained from Waste Eggshells and Used Lithium-ion Batteries as a Sustainable Development Approach

Manuscript ID: nanomaterials-3079115

Authors would like to thank the Editor and Reviewers for their insightful queries and suggestions. Based on the reviewer queries the manuscript has been revised incorporating all the required contents. The author responses are highlighted in red color   both in author’s response and manuscripts.

Reviewer-2

Reviewer report on manuscript nanomaterials-3079115

Kathalingam Adaikalam et al. “Energy Storage Application of CaO/Graphite Nanocomposite Powder obtained from Waste Eggshells and Used Lithium-ion Batteries as a Sustainable Development Approach”

 This study reports the conversion of waste eggshell into calcium oxide by high-temperature calcination and extraction of nanographite from spent batteries for application in energy storage fields. Both CaO and CaO/graphite were characterized for their structural, morphological, and chemical compositions using XRD, SEM, TEM, and XPS techniques. The prepared CaO/graphite nanocomposite material was evaluated for its efficiency in electrochemical supercapacitor applications. CaO and its composite with graphite powder obtained from used lithium-ion batteries demonstrated improved performance compared to CaO alone for energy storage applications.   

 The manuscript can be accepted only after major revision.

Below, I point out several questions to help the authors improve the manuscript before publication.

Questions/comments:

Reviewer query

  1. The introduction doesn’t provide sufficient background and doesn’t include all relevant references, e.g.  [Yalovega et al. Interfacial interaction in NiOx/MWNTs and CuOx/MWNTs nanostructures as efficient electrode materials for high-performance supercapacitors.Nanomaterials, 2024, 14(11), 947] and [Shmatko et al. Interaction between NiOx and MWNT in NiOx/MWNTs composite: XANES and XPS study. Journal of Electron Spectroscopy and Related Phenomena, 2017, 220, 76-80].

Author Response

As suggested, the introduction part has been improved adding more relevant points referring the mentioned reports.

Reviewer query

  1. More details to the section “2. Experimental details” should be added.

Author Response

The experimental part has been revised giving full details of the instruments used and procedures.

Reviewer query

  1. Detailed information about X-ray photoelectron spectra recording should be added. The energy resolution and pass energy values during X-ray photoelectron spectra acquisition should be written. The procedures for background subtraction and spectra deconvolution procedure should be well described.

Author Response

The parameters used to record the XPS spectrum is given below for reviewer’s information, it has suitably been included in the revised manuscript. For deconvolution of the peaks Fityk software was used adapting its procedures.

Reviewer query

  1. It is well-known that sp2component of the X-ray photoelectron spectra has an asymmetric shape from the side of higher binding energies, which is described by the Doniach-Sunjic function, which should be used for fitting of the sp2 component in the C1s spectrum. As an example, I recommend using the publication [J. Alloys Compd. 2019, 792, 713-720].

Author Response

The manuscript has been revised referring the suggested articles and more related works. The C1s spectrum also re-prepared as per the instruction, and included in the revised manuscript, and according the text of the manuscripts also changed.

Reviewer query

  1. The XPS spectra fitting is not very good justified. There are not up-to-date references for choice of the components. I recommend using the publications [Small, 2023, 19, 2208265] and [Appl. Surf. Sci. 2022, 590, 153055].

Author Response

As per the suggestion the XPS has been modified and included in the revised manuscript referring the works related.

Reviewer query

  1. There are errors in the interpretation of peaks in the X-ray photoelectron spectra that must be corrected before the manuscript publication. I recommend referring to the publications [Small, 2023, 19, 2208265], [Appl. Surf. Sci. 2022, 590, 153055], and [Carbon, 2022, 194, 52-61].

Author Response

Thanks to the reviewer for his suggestions to improve the manuscript, and as suggested the XPS result has thoroughly been corrected in the revised manuscript.

Reviewer query

  1. The statement “Figure 7(d) shows the fitted C 1s spectra with peaks at 285 eV, 286 eV, and 288 eV, assigned to C–C, C–O, and C=O vibrations, respectively, due to the association of carbon with CaO” (page 8) is not correctly for X-ray photoelectron spectra and must be corrected. For corrections, please, refer to the mentioned above publications.

Author Response

It is corrected after re-drawing the XPS plots and adding more points

Reviewer query

  1. It’s not clear how Authors made the conclusion “This XPS result also confirms the combination of CaO and graphite in the prepared composite material.” (page 8). This discussion should be extended.

Author Response

In this work, we have prepared CaO/graphite nanocomposite using CaO and graphite powders obtained from waste eggshells and graphite rods obtained from spent batteries. From the characterizations such as XRD, SEM, TEM and XPS done on the prepared composite and the elements presented made as to come to the conclusion.  As reviewer claimed, it is not scientifically authenticate. So, in the revised version we have corrected including reasonable conclusions.

Reviewer query

  1. Section “Funding” is missed.
  2. Section “Author Contributions” is missed.
  3. “Data availability statement” is missed.
  4. “Acknowledgments” is missed.
  5. “Statement about conflict of interest” is missed.

Author Response

All about “Funding”, “Author Contributions”, “Data availability statement”, “Acknowledgments” and “Statement about conflict of interest” are mentioned in the revised manuscript.

Reviewer query

References should be corrected and reformatted. Almost all references have missing title of journals, e.g.  [1], [2], [7], [14-16], [19], [20], [22], [23], [25-29], and [31-33].

Author Response

Sorry for this mistakes, actually reference was done using EndNote that is why we did not checked and noticed it. All the references are now checked and corrected to have all the necessary information.

Round 2

Reviewer 1 Report

Comments and Suggestions for Authors

All suggestions have been well revised, and it could be accepted in present form. 

Author Response

Thanks for the reviewer for his positive response

Reviewer 2 Report

Comments and Suggestions for Authors

Reviewer report on manuscript nanomaterials-3079115_Revision-1

Kathalingam Adaikalam et al.Energy Storage Application of CaO/Graphite Nanocomposite Powder obtained from Waste Eggshells and Used Lithium-ion Batteries as a Sustainable Development Approach

This study reports the conversion of waste eggshell into calcium oxide by high-temperature calcination and extraction of nanographite from spent batteries for application in energy storage fields.   

The manuscript can be accepted after minor revision.

Authors should make the small technical correction in the manuscript before publication.

Comment:

1.      References [5] and [36] are the same, namely, [Yalovega et al. Nanomaterials, 2024]. It seems that one of these references should be replaced by the reference [Shmatko et al. Interaction between NiOx and MWNT in NiOx/MWNTs composite: XANES and XPS study. Journal of Electron Spectroscopy and Related Phenomena, 2017, 220, 76-80].

Author Response

Reviewer comment

  1. References [5] and [36] are the same, namely, [Yalovega et al. Nanomaterials, 2024]. It seems that one of these references should be replaced by the reference [Shmatko et al. Interaction between NiOxand MWNT in NiOx/MWNTs composite: XANES and XPS study. Journal of Electron Spectroscopy and Related Phenomena, 2017, 220, 76-80].

Author Response

Thank you very much for the identification of the mistakes, it is corrected in the revised manuscript.